# Relationship between Inflammatory Food Consumption and Age-Related Hearing Loss in a Prospective Observational Cohort: Results from the Salus in Apulia Study

**DOI:** 10.3390/nu12020426

**Published:** 2020-02-07

**Authors:** Rodolfo Sardone, Luisa Lampignano, Vito Guerra, Roberta Zupo, Rossella Donghia, Fabio Castellana, Petronilla Battista, Ilaria Bortone, Filippo Procino, Marco Castellana, Andrea Passantino, Roberta Rucco, Madia Lozupone, Davide Seripa, Francesco Panza, Giovanni De Pergola, Gianluigi Giannelli, Giancarlo Logroscino, Heiner Boeing, Nicola Quaranta

**Affiliations:** 1Frailty Phenotypes Research Unit, “Salus in Apulia Study”, National Institute of Gastroenterology “Saverio de Bellis”, Research Hospital, Castellana Grotte, 70013 Bari, Italyf_panza@hotmail.com (F.P.);; 2Data Analysis Unit, National Institute of Gastroenterology “Saverio de Bellis”, Research Hospital, Castellana Grotte, 70013 Bari, Italy; vito.guerra@irccsdebellis.it (V.G.);; 3Department of Cardiology and Cardiac Rehabilitation, Scientific Clinical Institutes Maugeri, IRCCS Institute of Bari, 70124 Bari, Italy; 4Center for Neurodegenerative Diseases and the Aging Brain, University of Bari Aldo Moro, 70100 Bari, Italy; 5Research Laboratory, Complex Structure of Geriatrics, Department of Medical Sciences, Fondazione IRCCS Casa Sollievo della Sofferenza, San Giovanni Rotondo, 71013 Foggia, Italy; 6Clinical Nutrition Unit, Medical Oncology, Department of Biomedical Science and Human Oncology, University of Bari Aldo Moro, School of Medicine, 70100 Bari, Italy; 7Department of Clinical Research in Neurology, “Pia Fondazione Cardinale G. Panico”, Tricase, 73039 Lecce, Italy; 8German Institute of Human Nutrition Potsdam-Rehbrücke, 14558 Nuthetal, Germany; 9Otolaryngology Unit, Department of Basic Medical Science, Neuroscience and Sense Organs, University of Bari Aldo Moro, 70100 Bari, Italy

**Keywords:** age-related hearing loss, presbycusis, population studies, diet, food, sugar, alcohol, inflammation

## Abstract

Age related hearing loss (ARHL) affects about one third of the elderly population. It is suggested that the senescence of the hair cells could be modulated by inflammation. Thus, intake of anti- and pro-inflammatory foods is of high interest. Methods: From the MICOL study population, 734 participants were selected that participated in the 2013 to 2018 examination including hearing ability and from which past data collected in 2005/2008 was available. ARHL status was determined and compared cross-sectionally and retrospectively according to clinical and lifestyle data including food and micronutrient intake. Results: ARHL status was associated with higher age but not with education, smoking, relative weight (BMI), and clinical-chemical blood markers in the crossectional and retrospective analyses. Higher intake of fruit juices among ARHL-participants was seen cross-sectionally, and of sugary foods, high-caloric drinks, beer, and spirits retrospectively. No difference was found for the other 26 food groups and for dietary micronutrients with the exception of past vitamin A, which was higher among normal hearing subjects. Conclusions: Pro-inflammatory foods with a high-sugar content and also beer and spirits were found to be assocated with positive ARHL-status, but not anti-inflammatory foods. Diet could be a candidate for lifestyle advice for the prevention of ARHL.

## 1. Introduction

Age-related hearing loss (ARHL) is an important and prevalent disease in the elderly population. According to the World Health Organization (WHO), about 18% of the population over 65 years are affected by disabling hearing loss [1]. The WHO defines ARHL as a loss of hearing ability of more than 40 dB [1] in the better ear, measured by pure-tone audiometry. 

There is still ongoing debate about the origin and the context of ARHL. In the last two decades a link between age-related hearing loss (ARHL) and dementia and other age-related outcomes has been well established [2]. Thus, ARHL could be considered as a good marker of the aging process. The context makes it difficult to establish risk factors for this specific outcome in view of the many facets of the aging process and the factors driving or slowing down the process. Therefore, it is not surprising that only a few population studies in the literature have described the factors related to the incidence or prevalence of ARHL [2]. The establishment of such factors - particularly if modifiable—with sufficient evidence could be an important step forward towards the prevention of ARHL and could potentially allow the repair of early damage to the hearing before it can reach an irreversible state [3]. Such investigations also need to be seen as a general public health approach not only to understanding and preventing hearing loss but also to tracking the long-term trajectories of age-related chronic diseases.

The etiology of age-related hearing impairment is highly heterogeneous, multifactorial and much debated in modern science. A major hypothesis about aging, and subsequently hearing loss, regards inflammation [4]. It had already been speculated that inflammation could have a fundamental role in the onset and degeneration of cochlear damage [5]. In recent years there has been accumulating evidence linking age-related hearing loss with numerous aging disorders such as cerebrovascular diseases (CVDs) [6] and physical frailty [7]. The prevalence of both age-related hearing loss and CVD is increased in older adults [8,9] and these conditions often coexist, suggesting that there may be some common underlying factors [6]. It is well known that serum markers of systemic inflammation increase with age and have been associated with cardiovascular outcomes and all-cause mortality [10,11,12,13]. Preventing systemic inflammation and microcirculation abnormalities could therefore be a public health strategy to lower the incidence of ARHL [2]. It is well known that systemic inflammation could be decreased or increased by the consumption of peculiar foods [14] and by specific dietary patterns [15]. Anti-inflammatory foods rich in Vitamin A, C, and E like vegetables, fresh fruits and nuts has been considered to be the most important foods protective of ARHL [16]. Likewise, pro-inflammatory foods, like sugar-rich juices, desserts and alcoholic drinks have been shown to increase the risk of onset of ARHL [17]. The relationship between inflammation processes and age-related hearing loss is well depicted in the biology of inner ear damage both in animal models and in population studies [5,18]. Despite this evidence, the association between particular foods or diet is still unclear [16].

The aim of this study was therefore to investigate which foods were associated with a presence of ARHL cross-sectionally and retrospectively in a population-based study on elderly subjects located in Southern Italy.

## 2. Methods

### 2.1. Study Population

Participants of the present study were recruited from the electoral rolls of Castellana Grotte, Bari, Southern Italy, within the MICOL studies (*n* = 2472) and the GreatAGE Study (*n* = 2526) [19,20,21]. The baseline data (MICOL3, M3) were recorded from 2003 to 2005 and the follow-up data from 2013 to 2015 (GreatAGE Study -MICOL4, M4). The GreatAGE study has been described elsewhere [20,21,22,23]. All the studies were conducted within the “Salus in Apulia Study”, a public health initiative funded by the Italian Ministry of Health and Apulia Regional Government and conducted at IRCCS “S. De Bellis” Research Hospital. For the M4 study, residents ≥ 65 years from the general population of Castella Grotte were invited. The invitation included also subjects of the MICOL studies that were in the respective age range above 64 years. In the M4 examination, in addition to the assessment of clinical and lifestyle aspects, a hearing assessment was also performed (see Section 2.2, Hearing Assessment). For this study we used only the data from those M4 study participants that already participated in the M3 examination (*n* = 734). This selection of study participants allowed to utilize past data for the investigation. All participants signed informed consent before their examination and the study was approved by the IRB of the head institution, the National Institute of Gastroenterology and Research Hospital “S. de Bellis” in Castellana Grotte, Italy. The studies are in accord with the Helsinki Declaration of 1975. The present study adhered to the “Standards for Reporting Diagnostic Accuracy Studies” (STARD) guidelines (http://www.stard-statement.org/) and the manuscript was organized according to the the “Strengthening the Reporting of Observational Studies in Epidemiology-Nutritional Epidemiology” (STROBE-nut) guidelines (https://www.strobe-nut.org/).

### 2.2. Hearing Assessment

The examination was performed by a qualified audiologist according to Italian law following the international standard ISO 8253-1:2010. All participants underwent tympanometry, and stapedial reflexes (Clarinet Plus, Middle Ear Analyzer, Inventis, Italy) were checked to exclude middle and external ear disorders that could induce conductive hearing loss. Peripheral ARHL was defined as a pure tone average (PTA) threshold higher than 40 dB hearing level (HL) in the better ear according to the World Health Organization (WHO) definition of disabling hearing loss [1], assessed with pure tone audiometry, following the Hughson–Westlake method, in a soundproof booth with HDR 39 headphones (Sennheiser electronic GmbH & Co. KG, Wedemark, Germany) and PIANO Audiometer (Inventis SRL, Padova, Italy), calibrated and executed according to international standards for audiometric testing. The PTA was calculated at the frequencies of 0.5, 1. and 2 KHz. 

### 2.3. Dietary Assessments

Diet was assessed with the food frequency method applied in both examinations. The selfadministered Food Frequency Questionnaire (FFQ) was structured in eleven sections that partly mirror the sequence of food intake during the day and include foods of similar characteristics: grains, meat, fish, milk and dairy products, vegetables, legumes, fruits, miscellaneous foods, water and alcoholic beverages, olive oil and other edible fats, coffee/sugar and salt. In a further step, the FFQ was validated against dietary records and the results were reviewed to make any necessary modifications of the questionnaire [24]. In the final questionnaire, 85 food items were considered to best reflect the regional diet, together with some questions about the use of edible fats. The latter were not quantified but summarized in a separate food group (19 edible cooking fats). The 85 food items of the FFQ and the questions about the use of fat were regrouped and shortened by number and also made concordant with a study representing a Spanish (Mediterranean) diet including 26 food groups from an original group of 117 food items [25]. For our FFQ questionnaire, we established 30 food groups of foods of a similar type. Twenty-four of them were identical to those in the Spanish study. One food group (19 edible cooking fats) could not be quantified and was not used in the present study. The establishment of the 29 food groups (30 minus the 1 non quantified food group) was also done to consent direct comparisons with other studies from the Mediterranean area. Appendix A shows the concordance of the single foods in the questionnaire and the food grouping used in the analyses.

### 2.4. Micronutrients Assessment

Total energy and micronutrients intakes of sodium (Na), potassium (K), iron (Fe), calcium (Ca), phosphorus (P), thiamine (B1), riboflavin (B2), niacin (PP), and Vitamin A and C were calculated using Italian food composition tables [26].

### 2.5. Socioeconomic and Lifestyle Assessment

The examination included an interview and a questionnaire which covered socioeconomic and lifestyle variables such as years of schooling and smoking behaviour, For the study, the years of schooling variable was categorized following the the Italian national education system and formed the education variable in the analysis. The lowest level: < 6 years reflected primary school education, the middle level of 6–8 years reflected lower secondary school education, and the highest level of > 8 years reflected upper secondary school/high school education and university education. Smoking was assessed with the single question “Are you a current smoker?”, and categized as yes and no.

### 2.6. Clinical Characteristics

Blood samples were collected from the subjects in the morning after an overnight fast, and among other parameters, fasting blood glucose, total cholesterol, and triglycerides were measured, using standard automated enzymatic colorimetric methods (AutoMate 2550, Beckman Coulter, Brea, CA, USA), under strict quality control. The prevalence of diabetes mellitus was calculated on the basis of a diagnosis of diabetes given at the interview, the use of antidiabetic medications, and fasting blood glucose above 126 (mg/dL). A sphygmomanometer (YTON) and a stethoscope (FARMAC-ZARBAN) were used to measure blood pressure, by professional nurses with a professional qualification in Italy. Blood pressure was determined in a sitting position after rest. The final values of blood pressure (systolic and diastolic blood pressure) were the mean of the last two of three measurements. Height and weight measurements were performed using a Seca 220 altimeter and a Seca 711 scale. Body mass index (BMI) was calculated as kg/m^2^. Multimorbidity status was defined as the co-presence of two or more chronic diseases among the following pathological conditions: diabetes, hypertension, peptic ulcer, cholangiolithiasis, myocardial infarction, hepatic cirrhosis or other liver diseases, inflammatory bowel diseases, major infectious diseases, leukemia or other blood chronic diseases, viral hepatitis, AIDS [27]. All pathologies were assessed using a general anamnestic questionnaire administered by an expert physician.

### 2.7. Statistical Analysis

The characteristics of participants were reported as mean and standard deviation (M ± SD) for continuous variables, and as frequencies and percentages (%) for categorical variables. For the analysis, cross-sectional (M4 with ARHL measurements) as well as retrospective data (M3 examination reflecting the time period 12 years ago) were used. All subjects were subdivided into the ARHL (Yes/No) categories and compared at each time period according to mean (quantitative variable) or percentage (frequency variable) *P*-values of differences between the categories were adjusted for age, gender, smoking, education, BMI, diabetes mellitus, antihypertensive and statin drug assumption (asked with a single question) using logistic regression models. The exact adjustments for each logistic regression model were reported in the notes of the tables. All foods and food groups were calculated on quantity of daily consumption. Energy-adjusted nutrient intakes were computed as the residual from the regression model, with total energy intake as the independent variable and absolute nutrient intake as the dependent variable [28]. A *p*-value was considered significant if < 0.05. Formal analyses were done with STATA 16.0, StataCorp software. 2019. *Stata Statistical Software: Release 16*. College Station, TX: StataCorp LLC. 

## 3. Results

The participants (*n* = 734) examined in M3 and M4 were slighly dominated by males (57.90%). Mean age and education at M3 were 65.95 ± 6.59, and 73.54 ± 6.61 at M4; mean years of education were 6.63 ± 3.56 at M3 and 9.52 ± 4.20 at M4. It appears that in all examinations, the ARHL participants were older with about 4.5 years difference at each examination (Table 1). Otherwise, the ARHL subjects did not show differences compared with the non-ARHL counterparts in respect to education, and medical conditions or clinical-chemical blood measurements (Table 1). 

Table 2 is showing the results regarding food groups. The cross-sectional analysis (M4) revealed only the food group fruit juice that differed between the ARHL proups with a higher intake at the M4 examination (10.02 ± 5.92 in the ARHL group vs. 5.64 ± 2.03 in the normal hearing group). A significantly increased risk of ARHL was also evident for each unitary increase (mL) of juice consumption (*p* < 0.05). The retrospective analysis revealed more food groups than the cross-sectional analysis as being related to ARHL. We found a statistically significant mean differences in terms of an increased consumption of sugary foods (15.23 ± 34 in the ARHL group vs. 11.77 ± 1.96 in the normal hearing group), caloric drinks (16.21 ± 16.16 in the ARHL group vs. 9.21 ± 5.57 in the normal hearing group), beer (39.68 ± 42.23 in the ARHL group vs. 27.27 ± 23.92 in the normal hearing group) and spirits (2.52 ±2.93 in the ARHL group vs. 1.51 ± 1.38 in the normal hearing group) in the ARHL group compared to the normal hearing group. There were no other significant relationships between anti-inflammatory foods (leafy foods, nuts, olive oil, legumes) and ARHL. To further investigate the anti-inflammatory potential of diet, we investigated the micro-nutrients components compared among the ARHL groups crossectionally and restrospectively (Table 3). This analysis revealed that participants with non-hearing loss had a higher intale of dietary vitamin A 12 years before the hearing examination (1177.66 ± 99.71 vs 1042.04 ± 116.58).

## 4. Discussion

In the present study, we could link pro-inflammatory food groups and some alcoholic beverages with ARHL but not anti-inflamatory food groups. Further, ARHL clearly appeared to be age related but was not linked to classical age related medical and clinical-chemical blood conditions in this study. 

Sugary foods, alcohol, and caloric drinks are typically considered paradigms of an unhealthy lifestyle, not only related to diet, but also to smoking and sedentary behavior [29]. Unhealthy foods may interact with hearing thresholds in many altered pathways, previously well studied in animal models, involving increased inflammation linked to impairments of insulin signaling, abnormal proteostasis, oxidative stress and alterations of the intermediary metabolism in the inner ear [30].

In the present study, there was no significant relationship between smoking habit and ARHL, even though several studies have reported an association between hearing loss and smoking [31,32]. In particular, a meta-analysis suggested moderate to large associations between smoking and hearing loss [33]. Our finding could probably be related to the assessment method we used for smoking habit, taking into account only the current smoking status. 

Furthermore, in our study sample, the ARHL group had higher, but not significant, mean blood glucose levels, with a slightly higher prevalence of diabetes mellitus. Angiopathy, neuropathy, oxidative stress, and remnants of glycation end products may be the underlying mechanisms linking type 2 diabetes mellitus to ARHL [34]. A relationship between these two chronic conditions has been confirmed in two different animal model studies [35,36].

ARHL pathophysiology is classically caused by a combination of microvascular and neurodegenerative processes. Early evidence of ARHL physiopathology was suggested by the autoptic studies of Schuknecht and Gacek, who described findings in the human inner ear and compared these to premortem hearing tests [37]. Although the histological technique was crude compared with today’s standards, these studies provided the opportunity to characterize the hearing in temporal bones with a single pattern of cellular loss. After these seminal studies, two principal pathways have been suggested to explain the development of ARHL: abnormalities in the metabolism of K+ Na+ regulation in the inner ear and the breakdown of the blood–labyrinth barrier due to inflammatory processes [38]. The first hypothesis has been confirmed especially in animal model studies of induced hearing loss, showing degeneration of the stria vascularis, the highest density area of Na-K pumps in the inner ear [39,40]. The stria vascularis is heavily vascularised and has an extremely high metabolic rate. Histopathological studies of aging gerbils have provided strong evidence for vascular involvement in ARHL [41]. Morphometric analyses of lateral wall preparations contrast-stained to highlight blood vessels have shown losses in the strial capillary area in aged animals [41,42]. These findings clearly support the role of microcirculation alterations in ARHL pathways. Moreover, diabetes mellitus and glycemic abnormalities could be related to alterations of microvessel metabolism and may be considered the prominent risk factor for age-related conditions such as ARHL or macular abnormalities [43]. However, we could not find such relations in our study.

The inflammation hypothesis motivated us to start a systematic analysis regarding the role of food intake in ARHL development. Anti-inflammatory foods are connected with a high antioxidant content and dietary vitamin intake [4]. Free radical formation in the inner ear is a key mechanism of hearing loss [44], causing vasoconstriction and therefore death of the inner ear cells. Subsequent reperfusion of cochlear cells further contributes to free radical formation and further cell death, similar to stroke mechanisms. Inflammation is a normal adaptive response aimed at restoring tissue functionality and homeostasis after infection or mechanical tissue injury that could have unintended negative consequences. Previous authors recognized the potentially important role of inflammation in causing age related hearing loss, but there is still not a well defined mechanistic hypothesis on the pathophysiological pathway [45]. The most recognized one could be related to the mixed effect of vasoactive function (that leads to micro-ischaemic events) and endothelial dysfunction (cell apoptosis induction) of peripheral inflammatory cytokines [46]. Those pathological events could affect the complex micro-circulation of cochlea, accelerating contextually the senescence of the cytoneural structures (inner hair cells) and the dysfunction of stria vascularis metabolism [46]. Generally, pro-inflammatory foods, especially sugars, are typically associated not only with an increase of systemic inflammation, but particularly with micro-vessels damage in terms of micro-ischemic events [47]. The intake of those harmful foods could catalyze those pathological vascular effects, shared with the pathophysiology of age related hearing loss. 

Antioxidants such as vitamins, which inhibit the formation of free radicals, may play a specific role in preventing and treating ARHL [4,48]. Therefore, several studies have reported a relationship between ARHL and vitamins A [49,50], C [51], and E in humans [52]. In the present study, ARHL subjects used on average a lower amount of foods that naturally contain more vitamins A, C, and E, i.e., vegetables, fruits and nuts, but the difference was not significant. Lately, the focus on anti-inflammatory diets has been increasing. These diets, of which the Mediterranean diet is the best example, are characterised by a higher intake of vegetables, fruits, whole grains, legumes, nuts, fish, lean meat, dairy, olive oil, and moderate alcohol consumption (red wine) and a very low intake of processed foods with a rich sugar, salt and saturated fat content. This makes the diet a source of high-quality FAs (i.e., omega-3 and omega-9), fibers and complex carbohydrates, vitamins and minerals [53]. One of the most powerful effects of these diets is to help in the prevention and treatment of non-communicable diseases, such as cardiovascular diseases, type 2 diabetes, hypertension and cancer [54]. For example, the protective mechanisms of increasing vegetable and fruit intake against cardiovascular disease include decreasing blood pressure, regulation of lipid metabolism and reducing oxidative stress and low-grade inflammation. The high content of antioxidants (flavonoids, vitamin C, Vitamin E, ß-carotene) reduces DNA damaging. The protective effects of fruit and vegetables intake toward type 2 diabetes mainly depend on their rich fiber content, that improves insulin sensitivity. A favorable effect of nuts on cardiovascular health is due to the unique composition of these in monounsaturated fatty acids and polyunsaturated fatty acids, fiber, magnesium, arginine and polyphenols. Possible effects include a reduction of low-grade inflammation, oxidative stress, endothelial dysfunction and an improvement of the lipid profile and insulin resistance [55]. In Figure 1 are listed the pro-inflammatory and anti-inflammatory foods, specified in accordance with the food groups used for the statistical analysis.

To make a more focused analysis of the effect of food components on age-related hearing loss, we estimated the average amount of micronutrients intake obtained through the diet. We found a significant decrease in the risk of ARHL in subjects with a higher intake of vitamin A but not of the other vitamins. This highlights two major points: (1) the micronutrients assessment method using the FFQ is not the most appropriate to evaluate this kind of information since it cannot directly estimate the presence of retinoic acid (Vitamin A); (2) retinoic acid could have several functions in protection against age-related diseases (including hearing loss), mainly due to its delaying effect on apoptotic processes [56] and role in the renewal of the inner hair cells [57]. 

Overall, to better understand the impact of pro-inflammatory foods on ARHL, further epidemiological findings are needed.

### 4.1. Sugar as a Risk Factor for Age-Related Hearing Loss

Sugars, juices and caloric drinks (such as soda or coke) are among the foods with the highest glycemic index (GI) and glycemic load (GL). Dietary GI is commonly used to characterize the postprandial blood glucose response to the consumption of carbohydrates. GL is the product of a food’s GI and total carbohydrate content and represents both the quantity and quality of carbohydrates. High GI and GL diets have been shown to be associated with an increased risk of coronary heart disease, stroke, and type 2 diabetes mellitus [38]. Cardiovascular diseases have been proposed as a potential risk factor for ARHL, and a few studies have analyzed the association between cardiovascular-related potential risk factors and hearing capacity, including hypertension, diabetes mellitus, smoking history, and coronary heart disease [38]. Therefore, it is reasonable to hypothesize that diets rich in sugar, sugary fruit juices and caloric drinks could also increase the risk of developing ARHL [17]. On the other hand, the relationship between carbohydrate consumption and ARHL is thought to be related not to carbohydrates themselves, but to serum triglyceride levels. Consequently, diets rich in carbohydrates, especially sugars and drinks with highly concentrated fructose, such as juices, may lead to high serum triglyceride levels, and hence, be expected to affect auditory function [58].

### 4.2. Alcohol and Age-Related Hearing Loss

ARHL could be influenced by alcohol consumption, involving several underlying mechanisms such as an impairment of the cochlear blood supply, with resulting hypoxia and ischemic damage, oxidative stress and associated mitochondrial dysfunction, a loss of neurosensory cochlear cells, and neurodegeneration of central auditory pathways [59,60]. However, a moderate alcohol intake may be protective of cochlear blood flow [61], promoting cytoprotective and anti-inflammatory mechanisms that strengthen cellular survival pathways, and directly enhance neuroprotective mechanisms that preserve hearing [62]. By contrast, alcohol intake may also adversely alter central processing of auditory information [59,63].

Otherwise, an increasing body of evidence suggests that long-term moderate alcohol intake may protect against ARHL [64,65]. Some cross-sectional analyses reported an inverse association between alcohol consumption and ARHL [66,67], although other reports did not confirm this protective effect [63,64]. In a prospective study of 870 men and women aged 49 and older, no association was observed between alcohol consumption and the 5-year incidence of measured hearing loss, although cross-sectional analysis demonstrated a significant protective association between the moderate consumption of alcohol and ARHL [64]. In a prospective study of 26,809 older men, no association between total alcohol consumption and the risk of self-reported hearing loss was found [68]. It is worth recalling that a high alcohol intake is associated with elevated plasma triglycerides, linked to cardiovascular disease [69] and, as already pointed out, to auditory function [58]. Hence, the influence of long-term alcohol consumption on ARHL remains unclear. Given that the relation between alcohol consumption and other health outcomes varies according to the kind of beverage drunk [70,71], in the present study we explored the association of wine, beer and spirits consumption with ARHL and our analysis showed that a higher consumption of beer and spirits significantly increased the ARHL risk. Wine may not be hazardous due to the presence of polyphenols, in particular resveratrol in red wine, known for their antioxidant properties [72].

### 4.3. Strengths and Limitations

To the best of our knowledge, the present is the first population-based study to investigate the relationship between inflammatory food intakes and ARHL, in particular using instrumental methods to assess hearing loss [17]. A particular strength of the study was the utilization of past data that allow to asses relations that could induce ARHL. However, the use of past data is not exactly identical with a longitudinal observation, because the subjects examined at M3 did not include a hearing examination. Otherwise, fortunately, we could use data of the dietary and the clinical laboratory assessments from the past M3 examination. A further limitation is the FFQ as assessment method of diet, since it is memory-based and considered to be prone to measurement error limiting the ability to study the diet-disease relationships [73]. Despite the reported limitation of this assessment method, FFQs remain the dietary assessment method most commonly used to study dietary patterns and population habits [74]. Moreover, the results and significance do not agree between the two populations (GreatAGE and M3) (despite are the same osbserverd subjects) because between the two observation periods the eating habits of the surveyed subjects have changed, probably due both to the advancement of the age and to the change in lifestyle and taste of the population in general. 

## 5. Conclusions

ARHL is one of the most common chronic health conditions in older age [75], and its implications on important age-related outcomes like dementia and frailty are now well defined. The good news is that, despite the disastrous consequences that can originate, it is still one of the most effectively modifiable risk factors. It seems clear that defining protective factors that could delay ARHL onset in older age could have a great impact on healthy aging. The present study suggests that diet may have an effective role in preventing ARHL. In particular, avoiding inflammatory foods like sugars and alcohol can be part of a healthy lifestyle protocol that primary care physicians could easily implement in their common practice in both affected subjects and those at risk for ARHL. These findings support the concept that avoiding the intake of pro-inflammatory foods could affect age-related diseases in a more effective way than relying on the protective effect of anti-inflammatory or anti-oxidant foods. To better understand the causal effect of a particular food intake on the development of ARHL, specifically longitudinal population-based studies with a large sample size and well conducted intervention studies with foods already identified to be related to ARHL are needed.

## Figures and Tables

**Figure 1 nutrients-12-00426-f001:**
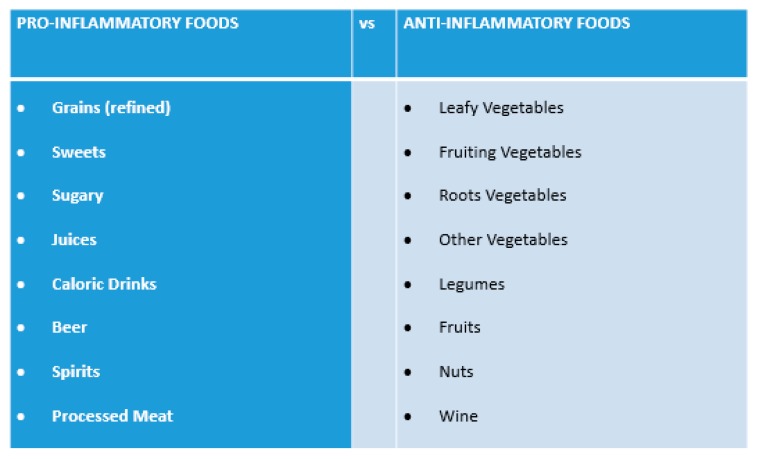
Classification of pro-inflammatory and anti-inflammatory foods, in accordance with the food groups used for the statistical analysis.

**Table 1 nutrients-12-00426-t001:** Sociodemographic and clinical characteristics of the population at baseline (M3) and follow-up (GreatAGE Study, M4) (*n* = 734).

	Great Age Study *(2012–18)*			MICOL 3 *(2005–06)*		
	Age-Related Hearing Loss (ARHL)			Age-Related Hearing Loss (ARHL)		
Variables *	No (< 40)	Yes (≥ 40)	*p* ^#^	No (< 40)	Yes (≥ 40)	*p* ^#^
Gender, men (%)	299 (60.53)	103 (57.87)	0.53	299 (60.53)	103 (57.87)	0.53
Age (years)	71.76 ± 5.67	76.18 ± 6.27	<0.0001	64.20 ± 5.68	68.76 ± 6.41	<0.0001
Smoke (%) ^(a)^	49 (10.01)	15 (8.54)	0.18	63 (12.91)	21 (11.36)	0.21
Education (%) ^(f)^						
Low	252 (51.48)	110 (61.80)	0.08	313 (63.84)	130 (73.84)	0.28
Medium	125 (25.48)	32 (17.98)	0.06	101 (20.64)	27 (15.34)	0.38
High	113 (23.03)	36 (20.22)	0.89	76 (15.51)	19 (10.82)	0.64
Diabetic (%) ^(a)^	96 (19.48)	39 (21.91)	0.87	89 (18.02)	32 (18.18)	0.33
Comorbidity (> 1) (%) ^(e)^	356 (72.39)	145 (81.46)	0.19	297 (60.12)	123 (69.32)	0.52
Systolic pressure (mmHg) ^(b)^	132.59 ± 3.88	134.04 ± 4.53	0.73	130.33 ± 6.23	134.16 ± 6.27	0.41
Diastolic pressure (mmHg) ^(b)^	79.05 ± 1.82	80.95 ± 6.56	0.68	75.67 ± 2.64	74.59 ± 3.08	0.56
BMI (kg/m^2^)	29.05 ± 4.96	29.00 ± 5.26	0.91	29.67 ± 4.73	29.82 ± 5.28	0.73
Glucose (mg/dL) ^(c)^	106.08 ± 17.56	109.35 ± 22.68	0.35	111.08 ± 21.17	111.35 ± 22.52	0.67
Total Cholesterol (mg/dL) ^(d)^	182.08 ± 17.47	179.70 ± 13.06	0.84	204.37 ± 10.10	198.87 ± 10.08	0.67
Triglycerides (mg/dL) ^(a)^	106.76 ± 1.81	110.34 ± 5.84	0.47	134.13 ± 13.12	142.14 ± 6.27	0.36

* Reported as: Mean and Standard Deviation (M ± SD). # *p*-value for logistic regression model (≥ 40 vs <40). Mean and logistic regression model adjusted for: (a) age, gender, and education; (b) age, gender, education, bmi, and anti-hypertetion drug; (c) age, gender, education, bmi, and diabetes; (d) age, gender, education, BMI, and statins drug; (e) age, gender, education, and bmi; (f) age, gender. MICOL 3 and Great-Age: All Foods Index and Food Groups were calculated on quantity daily consumption.

**Table 2 nutrients-12-00426-t002:** Dietary characteristics of the population at baseline (M3) and follow-up (GreatAGE Study, M4) (*n* = 734).

	Great Age Study *(2012–18)*			MICOL 3 *(2005–06)*		
	Age-Related Hearing Loss (ARHL)			Age-Related Hearing Loss (ARHL)		
Variables *	No (< 40)	Yes (≥ 40)	*p* ^#^	No (< 40)	Yes (≥ 40)	*p* ^#^
Food Groups ^(g)^						
Dairy	104.29 ± 10.97	94.75 ± 11.65	0.61	98.06 ± 8.16	88.78 ± 13.57	0.43
Low Fat Dairy	99.87 ± 6.96	96.67 ± 22.78	0.35	89.15 ± 8.97	104.72 ± 29.93	0.19
Eggs	7.65 ± 1.00	6.84 ± 1.54	0.40	7.67 ± 0.83	7.05 ± 1.42	0.82
White Meat	25.79 ± 4.34	24.69 ± 5.36	0.96	23.71 ± 3.85	22.50 ± 5.68	0.67
Red Meat	24.87 ± 5.71	24.26 ± 8.19	0.94	30.84 ± 5.98	30.17 ± 10.03	0.51
Processed Meat	15.48 ± 3.62	13.38 ± 4.74	0.80	16.90 ± 5.17	15.20 ± 7.26	0.56
Fish	26.85 ± 3.46	23.80 ± 5.28	0.62	26.69 ± 2.51	24.84 ± 14.68	0.60
Seafood/Shellfish	10.09 ± 0.95	10.18 ± 3.42	0.50	11.67 ± 1.69	10.99 ± 4.83	0.79
Leafy Vegetables	61.21 ± 6.84	56.28 ± 10.95	0.33	65.40 ± 13.13	61.95 ± 6.92	0.37
Fruiting Vegetables	94.56 ± 5.90	103.55 ± 14.96	0.33	101.92 ± 12.37	99.52 ± 11.20	0.71
Root Vegetables	11.38 ± 4.88	12.44 ± 5.55	0.58	8.84 ± 2.64	6.13 ± 0.99	0.15
Other Vegetables	87.58 ± 10.86	81.51 ± 19.71	0.58	81.82 ± 12.69	77.69 ± 8.48	0.42
Legumes	36.48 ± 3.11	40.38 ± 4.39	0.44	38.85 ± 5.57	39.73 ± 2.39	0.87
Potatoes	13.88 ± 2.55	16.82 ± 7.32	0.70	14.76 ± 2.24	16.61 ± 3.18	0.37
Fruits	625.65 ± 61.21	599.02 ± 115.68	0.59	662.72 ± 76.01	677.56 ± 105.27	0.52
Nuts	5.77 ± 1.67	4.89 ± 1.55	0.61	3.40 ± 0.61	4.01 ± 2.14	0.70
	**Great Age Study *(2012–18)***			**MICOL 3 *(2005–06)***		
	**Age-Related Hearing Loss (ARHL)**			**Age-Related Hearing Loss (ARHL)**		
Variables *	No (< 40)	Yes (≥ 40)	*p* ^#^	No (< 40)	Yes (≥ 40)	*p* ^#^
Grains	165.35 ± 23.05	175.14 ± 43.47	0.22	199.83 ± 32.83	207.72 ± 44.50	0.42
Olives and Vegetable Oil	53.84 ± 4.23	58.68 ± 14.03	0.29	47.72 ± 4.17	52.23 ± 7.50	0.54
Sweets	22.78 ± 2.90	21.53 ± 5.44	0.79	19.24 ± 3.02	20.78 ± 3.67	0.23
**Sugary**	10.35 ± 1.71	10.08 ± 1.30	0.72	**11.77 ± 1.96**	**15.23 ± 4.34**	**0.05**
**Juices**	**5.64 ± 2.03**	**10.02 ± 5.92**	**0.01**	12.44 ± 8.97	13.52 ± 13.00	0.15
**Caloric Drinks**	6.48 ± 6.58	6.91 ± 6.43	0.70	**9.21 ± 5.57**	**16.21 ± 16.16**	**0.05**
Ready to Eat Dish	31.07 ± 3.44	27.34 ± 8.09	0.76	34.96 ± 6.66	35.54 ± 11.64	0.13
Coffe	48.92 ± 9.00	43.21 ± 8.63	0.49	51.65 ± 10.92	48.86 ± 11.83	0.61
Wine	144.49 ± 72.36	142.47 ± 99.14	0.77	180.55 ± 100.13	176.59 ± 125.50	0.80
**Beer**	21.10 ± 17.22	17.61 ± 23.40	0.91	**27.27 ± 23.92**	**39.68 ± 42.23**	**0.02**
**Spirits**	1.26 ± 0.94	1.20 ± 1.14	0.66	**1.51 ± 1.38**	**2.52 ± 2.93**	**0.05**
Water	658.57 ± 21.69	684.00 ± 41.79	0.25	615.41 ± 46.84	618.56 ± 51.48	0.95

* Reported as: Mean and Standard Deviation (M ± SD). # *p*-value for logistic regression model (≥40 vs <40). Mean and logistic regression model adjusted for: (g) age, gender, smoking, education, and BMI. MICOL 3 and Great-Age: All Foods Index and Food Groups were calculated on quantity daily consumption. The statistically significant data are highlighted in bold.

**Table 3 nutrients-12-00426-t003:** Micronutrients intake characteristics of the population at baseline (M3) and follow-up (GreatAGE Study, M4) (*n* = 734).

	Great Age Study *(2012–18)*			MICOL 3 *(2005–06)*		
	Age-Related Hearing Loss (ARHL)			Age-Related Hearing Loss (ARHL)		
Variables ^§^	No (< 40)	Yes (≥ 40)	*p* ^#^	No (< 40)	Yes (≥ 40)	*p* ^#^
*Micro* ^(g)^						
Na	1552.70 ± 76.03	1522.53 ± 77.14	0.87	1672.40 ± 74.03	1629.64 ± 116.91	0.86
K	3406.54 ± 156.21	3309.85 ± 273.14	0.30	3609.79 ± 233.66	3506.99 ± 279.83	0.51
Fe	11.29 ± 0.68	11.21 ± 0.90	0.76	12.29 ± 0.67	11.85 ± 0.81	0.20
Ca	877.94 ± 72.16	853.74 ± 135.23	0.23	902.87 ± 79.12	865.53 ± 121.96	0.19
*p*	1145.63 ± 52.95	1125.27 ± 80.17	0.41	1225.24 ± 41.57	1179.77 ± 74.49	0.10
B_1_	0.83 ± 0.04	0.82 ± 0.07	0.65	0.89 ± 0.04	0.85 ± 0.05	0.36
B_2_	1.42 ± 0.08	1.42 ± 0.19	0.97	1.48 ± 0.08	1.44 ± 0.11	0.32
PP	1.42 ± 0.08	1.42 ± 0.19	0.97	1.48 ± 0.08	1.44 ± 0.11	0.32
Vit.A	1103.65 ± 135.87	1407.29 ± 698.70	0.11	**1177.66 ± 99.71**	**1042.04 ± 116.58**	**0.04**
Vit.C	183.94 ± 16.44	184.95 ± 21.25	0.65	189.24 ± 22.22	180.75 ± 30.20	0.34

# *p*-value for logistic regression model (≥ 40 vs <40). ^§^ Nutrients adjusted for calories. Mean and logistic regression model adjusted for: (g) age, gender, smoking, education, and BMI. sodium (Na), potassium (K), iron (Fe), calcium (Ca), phosphorus (P), thiamine (B1), riboflavin (B2), niacin (PP) MICOL 3 and Great-Age: All Foods Index and Food Groups were calculated on quantity of daily consumption. The statistically significant data are highlighted in bold.

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
