# Peer review of "Relationship between Inflammatory Food Consumption and Age-Related Hearing Loss in a Prospective Observational Cohort: Results from the Salus in Apulia Study"

_nutrients, 2020, doi:10.3390/nu12020426_

Round 1

Reviewer 1 Report

In this manuscript, the authors explored the hypothesis that pro-inflammatory foods  consumption could be associated with age-related hearing loss (ARHL) in a population-based study of elderly subjects in a Mediterranean area. This research topic is meaningful and positive results were obtained. I recommended accepting this article are major revision.

The results clearly proved that pro-inflammatory foods may have negative impacts on ARHL. However, this manuscript did not listed what foods are pro-inflammatory ones and what are not. A comprehensive table should be added to summary the foods involved in this study and classify them into pro-inflammatory and anti-inflammatory classes.

The possible reasons and mechanisms should be descriped why the certain pro-inflammatory foods may cause ARHL.

Reviewer 2 Report

In the sentence in lines 207-209, you use the same numbers to describe the difference between normal hearing and AHRL groups. Double check this.

In the discussion/limitations, I would discuss how you would explain why the two populations (GreatAge and MICOL) did not agree on the significance of association of certain variables. (eg. MICOL-3 had a significant association between sugary drinks and ARHL but the GreatAge did not.)

Round 2

Reviewer 1 Report

Most of the concerns have been issued. I recommend accepting this manuscript.